

# Unconscious priming shares a common resource pool with the manipulation subsystem

Xuechen Mao and Anmin Li

School of Psychology, Shanghai University of Sport, Shanghai, Shanghai, China

## ABSTRACT

**Background:** Working memory can be subdivided into two relatively independent subordinate systems, the maintenance subsystem and the manipulation subsystem. Although the two subsystems are quite heterogeneous, research thus far has not adequately distinguished the resource pools of the two subsystems. Additionally, previous research on the relationship between working memory and unconscious priming is paradoxical. Different subsystems leading to different effects on unconscious priming might be the reason for the paradoxical research. Therefore, the current article aimed to distinguish the resource pools among two working-memory subsystems and to investigate the relationship between the two subsystems and unconscious priming.

**Methods:** To address these issues, a maintenance dual-task and a manipulation dual-task program were developed. Each participant had to separately perform the two dual tasks in a balanced order. In each dual task, participants first completed a masked priming task accompanied by working-memory load. As a control, participants completed a prime identification test to confirm that the processing of the masked prime was at the unconscious level. The maintenance dual task comprised sandwich masking trials accompanied by Sternberg trials, while the manipulation dual task comprised sandwich masking trials accompanied by N-back trials.

**Results:** The results of the prime identification test indicated that the participants could not consciously perceive the masked prime of both dual tasks. The results of the working-memory task of both dual tasks indicated that the load manipulation was successful for both dual tasks. Most importantly, the results of the masking task of both dual tasks showed that an increase in working-memory load decreased the magnitude of unconscious priming in the manipulation dual task, whereas an increase in working-memory load did not decrease unconscious priming in the maintenance dual task. These observations demonstrate that the manipulation subsystem, rather than the maintenance subsystem, interferes with unconscious priming. Together with previous research, we propose a two-pool attention resource model to explain the modulation of working memory on unconscious priming by dissociating the executive resource pool of the manipulation system from the retention resource pool of the maintenance system. Thus, the current work confirms and extends the extant literature about the dependence of unconscious processing on

Corresponding author
Anmin Li, anminli@sus.edu.cn

attention resources by suggesting that unconscious priming shares a common resource pool with the manipulation subsystem.

## INTRODUCTION

According to a multicomponent model and its updates, working memory can be subdivided into two relatively independent subordinate systems (*Baddeley, 2012*; *Engle, 2018*; *Logie, Camos & Cowan, 2021*). One is the maintenance subsystem, which is responsible for the short-term maintenance of modality-specific information, and the other is the manipulation subsystem, which is the central executive and responsible for online monitoring, updating, and manipulation of information. Given that the limitation in working-memory resources chiefly originates in limited attention resources (*Chow & Conway, 2015*) and that attention has a variety of resource pools (*Cohen et al., 2012*), it is conceivable that the two working-memory subsystems share various resource pools. Although the two subsystems are quite heterogeneous, research thus far has not adequately distinguished the resource pools of the two subsystems (*Vaughan & Laborde, 2021*).

In view of the resource limitations of attention, when two tasks compete for a common resource pool, an increase in the workload of the secondary task leads to a decrease in the performance of the primary task (*Chun, Golomb & Turk-Browne, 2011*). Therefore, the resource pools of the two subsystems can be distinguished by introducing a special processing type that dedicates a resource pool to one of the working-memory subsystems without occupying the resource pool pf the other working-memory subsystem.

However, what is the suitable "special processing"? Previous research has found that unconscious priming depends on temporal attention (*Fabre, Lemaire & Grainger, 2007*; *Naccache, Blandin & Dehaene, 2002*). Furthermore, temporal attention has been confirmed to be modulated only with the manipulation subsystem (*Capizzi, Sanabria & Correa, 2012*) rather than the maintenance subsystem (*Zanto et al., 2020*). The dependence degree of unconscious priming on the resource pools of the two subsystems seems to be different. Therefore, whether unconscious priming is the appropriate "special processing" deserves exploring. To our knowledge, however, this topic has not yet been studied.

On the other hand, previous theories of the relationship between working memory and unconscious priming are paradoxical. Despite the controversy, in contrast to controlled processes, which are widely considered to be intentional, goal dependent, and conscious, automatic processes are generally considered to be unintentional, uncontrolled/ uncontrollable, goal independent, autonomous, purely stimulus driven, unconscious, efficient, and fast (*Moors & De Houwer, 2006*). Based on the automaticity view, unconscious priming is considered a kind of automatic process and therefore is assumed not be susceptible to attention resources. Research supporting this assumption reported no

observation of the modulation of working-memory load on unconscious priming (*Bodner & Stalinski, 2008*; *Perea et al., 2018*). Moreover, models of executive control consider unconscious priming to depend on the central executive. Research supporting this assumption reported that increased working-memory load decreases the processing of invisible stimuli (*Ansorge, Kunde & Kiefer, 2014*; *Bahrami, Lavie & Rees, 2007*; *Hung, Wu & Shimojo, 2020*).

Therefore, the second topic is the real relationship between working memory and unconscious priming. *Bodner & Stalinski (2008)* instructed one group to perform an unconscious identity priming task and another group to perform an unconscious identity priming task while maintaining several digits in mind. They found that there was no significant difference in unconscious priming between the two groups. Additionally, instead of a between-subject design, one study (*Perea et al., 2018*) employed a within-subject design and asked participants to perform an unconscious repetition priming task while maintaining four repeated consonants (low load), such as BBBB, or four different consonants (high load), such as BDKF. No changes were observed in unconscious priming between low and high loads. On the other hand, *Bahrami, Lavie & Rees (2007)* observed a decrease in the BOLD response elicited by invisible stimuli when participants performed a rapid serial visual presentation task from a high-load condition to a low-load condition. In addition, *Bayramova et al. (2021)* recently employed auditory stimuli with a dual-task paradigm in which an N-back task was interleaved with a flanker task. The authors found that the inhibition effect from the incongruent trials was decreased with higher loads (2- and 3-back) compared to low loads (0- and 1-back). The inhibition effect is considered as the relatively slower response time under incongruent trials compared with congruent trials and is based on the awareness of the incongruent and congruent stimuli. Therefore, based on *Bayramova et al. (2021)*, the decreased inhibition effect with higher loads means the decreased awareness of the incongruent and congruent stimuli with increasing working-memory load.

Although previous findings are mixed, further analysis indicates that the load manipulation tasks utilized in those studies seem to involve different working-memory subsystems; thus, this could account for the paradoxical findings. The studies finding no modulation of working-memory load on unconscious processing utilized the tasks that involved only maintenance processes, whereas the studies finding working-memory load decreased unconscious processing utilized tasks that involved manipulation processes. Thus, it is conceivable that the two subsystems have different effects on unconscious priming. However, to the best of our knowledge, this hypothesis has not been confirmed.

In this line of work, a dual-task paradigm was employed, in which an unconscious priming task (the primary task) was intermixed with a working-memory task (the secondary task). The sandwich masking paradigm (*Geng et al., 2020*; *Kiefer, 2019*; *Kiesel, Kunde & Hoffmann, 2008*), which has been widely utilized to measure unconscious processes, was chosen for the primary task. Typically, in this paradigm, participants are instructed to perform a target discrimination task, where the target is preceded by a masked prime, which is the same (congruent condition) or different from the target (incongruent condition) (*Eimer & Schlaghecken, 1998*). The prime is presented within a

very short time and is masked by a forward and a backward mask; therefore, the prime remains invisible to the participants. Regardless of the invisibility of the prime to the participants, the unconscious priming effect can be observed. The classic unconscious priming phenomenon is manifested in the faster responses to the target under congruent conditions than under incongruent conditions (*Klotz & Wolff, 1995*) or greater accuracy under congruent conditions than under incongruent conditions (*Klotz & Neumann, 1999*). Some studies hold that the differences in accuracy between congruent and incongruent conditions with different load conditions are not significant because of the ceiling effect (*Kiefer, 2019*; *Naccache, Blandin & Dehaene, 2002*). Therefore, the processing of the masked prime in the present study is measured by comparing the difference in response times (RTs) between congruent and incongruent conditions, which is the magnitude of the unconscious priming effect (*Geng et al., 2020*; *Kiefer, 2019*; *Naccache, Blandin & Dehaene, 2002*; *Perea et al., 2018*).

According to the taxonomy of conscious, preconscious, and subliminal processing (*Dehaene et al., 2006*), unconscious priming investigated in the present study is a type of subliminal processing (etymologically 'below the threshold'), which is defined as a condition of information inaccessibility because of the short presentation of the information as well as forward and backward masking. Based on the relationship between the masked prime and the target, unconscious priming can be divided into perceptual unconscious priming (in which the prime and target are similar in appearance under congruent conditions) and semantic unconscious priming (in which the prime and target are similar in semantics under congruent conditions). Previous research (*Fabre, Lemaire & Grainger, 2007*; *Perea et al., 2018*) held that in contrast to semantic unconscious priming, which is a higher level of unconscious processes, perceptual unconscious priming is a lower level of unconscious processes that do not require cognitive resources and are not affected by higher levels of processes such as working memory.

To distinguish the resource pools among two working-memory subsystems and to investigate the relationship between the two subsystems and unconscious priming, we ran two dual tasks: (1) the maintenance dual task explored the effect of the maintenance system on unconscious priming, and (2) the manipulation dual task explored the effect of the manipulation system on unconscious priming. We therefore employed Sternberg's item-recognition task as the secondary task in the maintenance dual task and deployed the N-back task as the secondary task in the manipulation dual task. The Sternberg task involves maintenance processes, wherein the participant is asked to retain a memory item consisting of several items during a delay and finally decide whether a cue item matches a member of the memory item or not (*Sternberg, 1969*; *Sternberg, 1975*). With increasing load in the Sternberg task, reaction time to the cue item increases (*Sternberg, 1969*; *Sternberg, 1975*). In contrast, the N-back task involves manipulation processes in addition to maintenance processes (*Bayramova et al., 2021*; *Kattner, 2021*; *Miller, Lundqvist & Bastos, 2018*) in presenting the participant with a sequence of memory items and asking them to indicate whether the current cue matches the one presented n steps earlier. Typically, the N-back task requires a stepwise increase in N (first 1-back, followed by 2-back, then 3-back). Previous research has demonstrated that unconscious processing can

be influenced when the stimuli utilized in the unconscious priming task share a common visual representation with the stimuli utilized in the working-memory task (*Ozimi & Repov, 2020*). To exclude this additional variable, we made the stimuli in the working-memory task independent of those in the sandwich masking task. That is, we selected geometrical shapes as the stimuli in the sandwich masking task and chose Arabic numbers as the stimuli in the working-memory task.

Based on the literature, we hypothesized that unconscious processing shares a common resource pool with the manipulation subsystem without consuming the resource pool of the maintenance subsystem. Therefore, we expected that unconscious priming would remain invariant with increasing maintenance workload in the maintenance dual task and would decrease with increasing manipulation workload in the manipulation dual task. To better understand the relationship between the two working-memory subsystems and unconscious priming, a two-pool attention resources model for working memory was assumed. In the hypothetical model, we propose the existence of a retention resource pool, which is engaged in maintenance processes and is not required by unconscious priming, and we propose the existence of an executive resource pool, which is engaged in manipulation processes and is required by unconscious priming. Consistent with previous research finding that the two subsystems rely on distinct mechanisms (*Baddeley, 2012*; *Engle, 2018*; *Logie, Camos & Cowan, 2021*; *Vaughan & Laborde, 2021*), we propose that the retention resource pool is relatively independent of the executive resource pool; therefore, one is required by unconscious priming, while the other is not.

## METHOD

### Participants

The within-subject design was deployed to increase the statistical power and control for additional variables, such as individual differences, in the present study. Based on prior research (*Hung, Wu & Shimojo, 2020*; *Kiefer, 2019*; *Meng et al., 2019*; *Xia et al., 2021*), power analysis (G*Power 3.1, $\alpha = 0.05$, power = 0.80, effect size = 0.25) showed that a minimum 16 volunteers needed to participate. Therefore, we included 25 students (mean age, 20.56 with a range of 18–25 years, 11 men) who participated in the present experiment. The Ethical Committee of Shanghai University of Sport approved the present experiment (No. 102772021RT020), and all participants provided written informed consent.

### Materials

Ten Arabic numbers from zero to nine (horizontal × vertical, $0.7° × 1.0°$) were adopted as stimuli for the working-memory tasks. In addition, five ellipses (varying from $4.0° × 1.1°$ to $4.0° × 1.9°$) and diamonds (varying from $1.1° × 4.0°$ to $1.9° × 4.0°$) were adopted as stimuli for the sandwich masking tasks (*Geng et al., 2020*). Notably, one ellipse ($4.0° × 1.5°$) and one diamond ($1.5° × 4.0°$) were considered primes, while the remaining geometries were considered targets. All stimuli were in white with a dark gray background.

Two pictures with many line patterns ($4.0° × 4.0°$) were considered forward and backward

masks (*Meng et al., 2019*). Moreover, these visual stimuli, with a resolution of 1,280 × 720 pixels, were displayed on a 19-inch color monitor (refresh rate: 60 Hz, frame duration: 16.67 ms), and participants were seated at an approximately 60 cm viewing distance. The experimental routines were programmed in E-Prime 2.0 software (*Schneider, Eschman & Zuccolotto, 2002*).

## Procedure

Each participant had to separately perform the maintenance dual task and the manipulation dual task in 2 days. The order of performing the two dual tasks was balanced between participants. That is, half of the participants performed the maintenance dual task on the first day, and the others performed the manipulation dual task on the first day. On each experimental day, the participants first completed three practice sessions, then completed a dual task (the maintenance dual task or the manipulation dual task), and finally performed a masked prime discrimination test. Three practice sessions consisted of 60 trials of the sandwich masking task, 60 trials of the working-memory task (30 trials for each load block), and six blocks of either of the dual tasks (maintenance: nine sandwich masking trials were intermixed with nine modulated Sternberg trials per block; manipulation: nine sandwich masking trials were intermixed with nine N-back trials per block). The maintenance dual task included 40 blocks, containing 360 Sternberg trials and 360 sandwich masking trials, and the manipulation dual-task included 40 blocks, containing 360 N-back trials and 360 sandwich masking trials. Each dual task was divided into two load conditions. The participants first performed the dual task with low load then with high load, or *vice versa*. The order of the load conditions was counterbalanced across participants. The stimuli in each task are presented in a pseudorandom order.

### Maintenance dual task

In each trial (Fig. 1), a yellow fixation was shown for 500 ms, replaced by a memory item that was presented for 2,000 ms. For the low-load condition, the memory item was a random number. Regarding the high-load condition, the memory item was a number sequence consisting of six random numbers with no repetition from zero to nine, with no more than two numbers in ascending or descending order. Based on the pilot experiment, the chosen presentation duration of the memory items allowed each participant to have sufficient time to remember all the numbers in the memory item. During the presentation time, the participants attempted to remember the memory item because each number included in the memory item would be compared with the following memory probes within one memory block.

Thereafter, one sandwich masking trial was presented, which started with a white fixation shown for 500 ms, followed by a 200-ms forward mask, a 33-ms prime and a 33-ms backward mask sequentially. Then, a target was shown, and the participants were asked to respond to the target with their right hands (left arrow for diamond and right arrow for ellipse). In half of the masking trials, the target was a diamond, and in the other half, the target was an ellipse. Additionally, half of the masking trials were under the congruent condition, while the other half were under the incongruent condition. The four
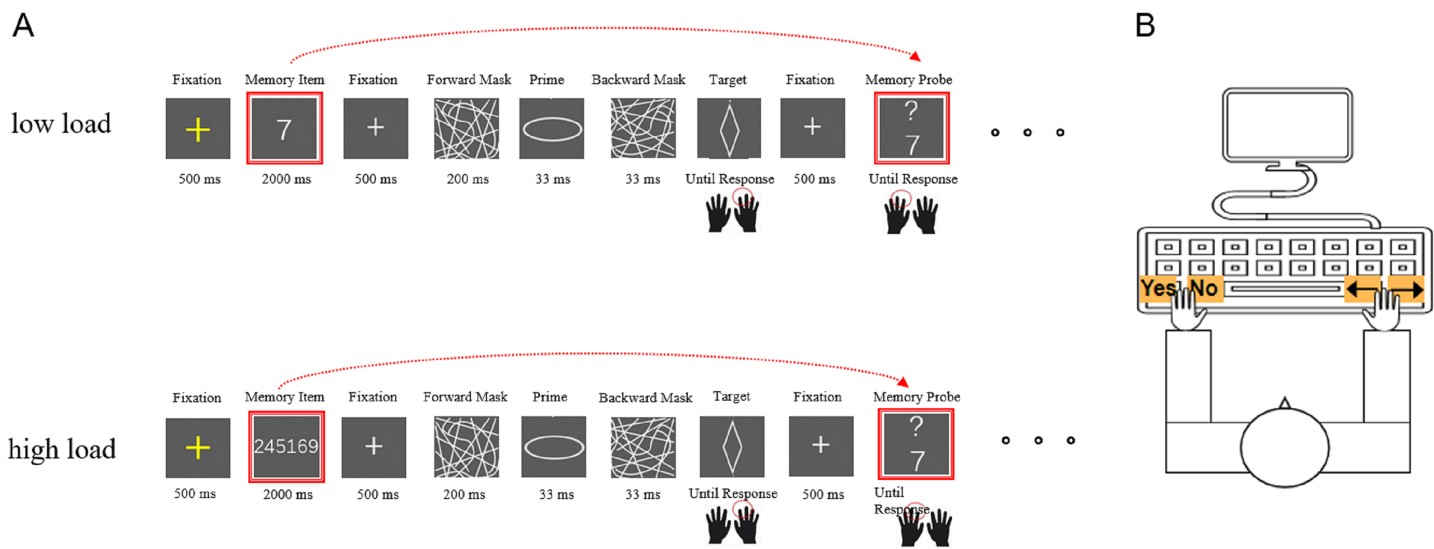

**Figure 1** **A schematic illustration of one block in the maintenance dual task.** (A) An example of one sandwich masking priming trial combined with a modulated Sternberg trial under low load (top) or under high load (bottom) in a maintenance dual task. (B) A schematic illustration of the key press during the dual-task paradigm.

diamond and four ellipse primes are shown with equal frequencies. The participants were instructed to respond as correctly and quickly as possible. After collecting a response, a white fixation was shown for 500 ms, followed by a memory probe, which was a single number with a question mark. The participants were instructed to respond to the probe with their left hands (press "yes" when the probe occurred in the memory item and press "no" when the probe did not occur in the memory item). Each number served equally often as the memory probe. In half of the memory trials, the probe occurred in the memory item, and in the other half, the probe did not. The participants were instructed to respond as correctly and quickly as possible. After collecting a response, the next masking trial was initiated.

*Manipulation dual task*

In each trial (Fig. 2) under low load, a yellow fixation was shown for 500 ms, replaced by a memory item that was presented for 2,000 ms. Thereafter, a white fixation was shown for 500 ms, indicating a sandwich masking trial was to come. The procedure of the masking trial was similar to that in the maintenance dual task. After completing the masking trial, a white fixation was shown for 500 ms, replaced by a memory probe, which was a single number with a question mark. The participants were instructed to respond to the probe with their left hands (press "yes" when the number probe was identical to the memory item and press "no" when the number probe was not identical to the memory item). Simultaneously, the participants attempted to remember the number probe because it would become the new memory item and was to be compared with a new number probe after completing one masking trial. Moreover, each number served equally often as the memory item. After collecting a response, the next masking trial was initiated.
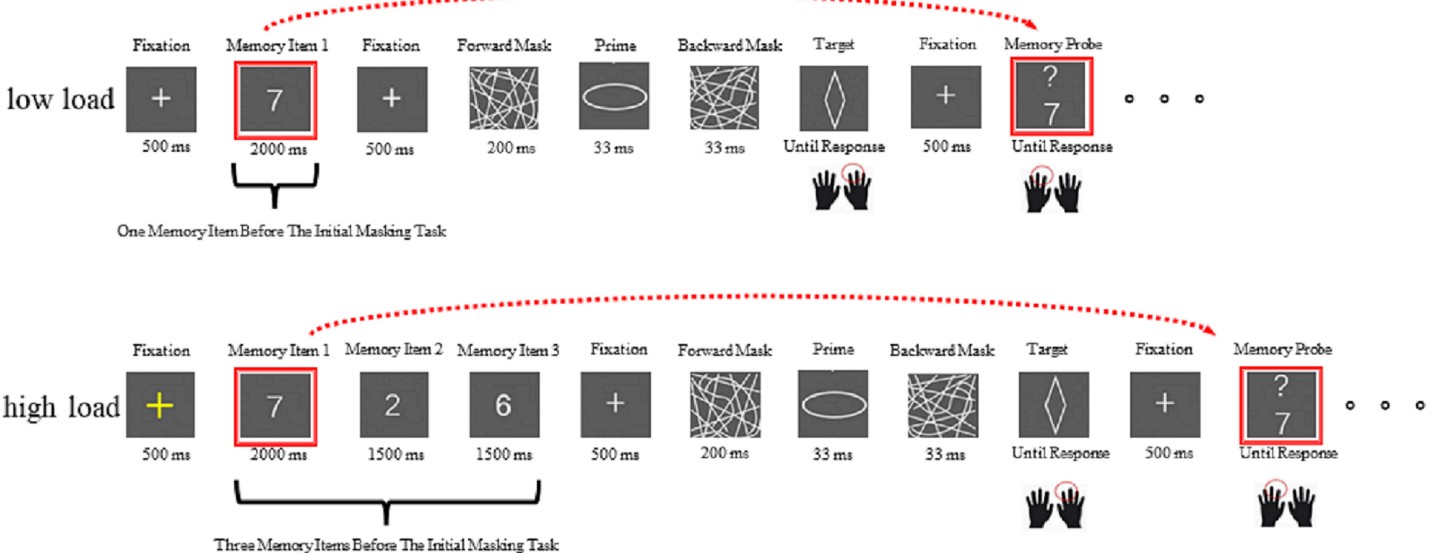

**Figure 2** A schematic illustration of one sandwich masking priming trial combined with an N-back trial under low load (top) or under high load (bottom) in the manipulation dual task.

The procedure of the high-load block was similar to that of the low-load block except (1) that after initial yellow fixation occurred, three memory items were presented sequentially, and each memory item was presented for 2,000 ms, and (2) that after the initial memory probe was displayed, the participants were instructed to decide whether the memory probe was identical to the first memory item and concurrently remembered this memory probe because this memory probe would become the new memory item and was to be compared with a new number probe after completing three masking tasks. That is, the initial first memory item leaves the set and is replaced by the current memory probe. In the next trial, the previous second item is probed. In summary, we employed the 3-back as the high-load condition and the 1-back as the low-load condition.

### Masked prime discrimination test

The procedure of each masked prime discrimination test was similar to the procedure of each dual task to keep the stimulation comparable (*Kiefer, 2019*). Only the low-load condition was used in the discrimination test because previous research suggested that the visibility of the masked prime was suppressed more effectively with high load than with low load (*Bahrami, Lavie & Rees, 2007*). We included both objective and subjective measures in the discrimination test to enhance its effectiveness (*Stein et al., 2021*). In each trial, in the objective measure, the participants were asked to identify the prime between the two masks and to perform the prime decision with the same response categories as in the main task. Following the button press, a short version of the 4-point perceptual awareness scale (PAS) was shown, which is the subjective measure. The participants had to choose one of the following options to rate the subjective visible levels of the masked prime: 1 = no experience, 2 = weak glimpse, 3 = almost clear, 4 = absolutely clear. Therefore, each discrimination block included objective and subjective measures for the

masked prime under low-load conditions. In total, the discrimination test contained 90 subjective and 90 objective trials, presented in 90 Sternberg trials after the maintenance dual task and 90 N-back trials after the manipulation dual task, respectively.

## Analysis

### Masked prime discrimination test

Based on previous research (*Geng et al., 2020*; *Kiefer, 2019*; *Meng et al., 2019*), the *d'* value was adopted to examine the visibility of the masked prime by calculating the hit rates (correct responses to the congruent prime) and false alarm rates (erroneous responses to the incongruent prime) from each participant's performance in the objective test. One-sample *t* tests were performed both on response accuracy (RA) of the objective test (tested against 0.5) and on *d'* (tested against 0). Data from participants whose RA of the objective test was significantly larger than chance (0.5) were excluded. Moreover, we compared the difference in the ratings of the subjective test, the RA of the objective test and *d'* between both dual tasks by paired *t* tests. Furthermore, Pearson's correlation between *d'* and unconscious priming under low load was performed. Finally, a two-tailed paired *t* test was separately performed on *d'*, the objective score and the subjective rating with two task conditions.

### Dual task

Based on prior research (*Kiefer, 2019*), individual responses with RTs above or below two standard deviations from the individual cell mean RT were discarded. The mean RT of the correct responses was entered into the analysis of the RT data. For the working-memory task, a two-tailed paired *t* test was separately performed on mean RT and RA with two load conditions for the maintenance dual task and the manipulation dual task. Furthermore, repeated-measures analyses of variance (ANOVAs) on mean RT and RA with the within-subject factors working-memory load (low *vs.* high) and working-memory subsystem (maintenance *vs.* manipulation) were performed. For the sandwich masking task, repeated-measures ANOVAs on mean RT and RA with the within-subject factors working-memory load (low *vs.* high) and prime congruency (congruent *vs.* incongruent) were performed for both dual-tasks. Moreover, repeated-measures ANOVAs on mean RT and RA with the within-subject factors working-memory load (low *vs.* high), prime congruency (congruent *vs.* incongruent) and dual-task type (the maintenance dual task *vs.* the manipulation dual task) were finally performed. Importantly, Greenhouse–Geisser corrections were applied when sphericity assumptions were violated, although these corrections did not change any inferences. Paired *t*-tests with LSD corrections for multiple comparisons were used for posthoc analysis if ANOVA showed significant interactions among different factors. To examine the order effects, the order variable was analyzed as a between-subject factor when conducting repeated-measures analyses of variance (ANOVAs) on mean RT. Bayes factors with JASP (*JASP Team, 2018*) were conducted to assess the likelihood when there was no interaction on RT and RA in the sandwich masking task as well as the working-memory task.

**Table 1 Performance of masked prime discrimination test in maintenance dual task and manipulation dual task.** Standard error in brackets.

| Masked prime discrimination test | Maintenance dual task | Manipulation dual task | 90% Confidence interval | |
|---|---|---|---|---|
| | | | Lower limit | Upper limit |
| Objective test (accuracy in percentage) | 53.2 (1.98) | 50.76 (1.88) | −0.03 | 0.08 |
| Subjective test (rating from 1 to 4) | 1.24 (0.06) | 1.4 (0.09) | −0.38 | 0.06 |
| D' of the Objective test | −0.06 (0.07) | 0.08 (0.06) | −0.36 | 0.08 |

## RESULTS

One participant's RA of the objective test was significantly larger than chance, which means the participant had an identification rate of the masked prime exceeding the 95% confidence (CI) interval of chance performance (*Kiefer, 2019*) and was therefore excluded from further analysis. Two participants' RA of the manipulation memory task under high load was more than two standard deviations below the group average (mean ± standard error: 95.31 ± 0.8%) and therefore were also excluded from further analysis. The remaining 22 participants were included in further analysis, which was sufficiently powered to detect the effects of interest (*Faul et al., 2007*).

### Masked prime discrimination test

Sandwich masking was effectual. There was no significant difference in masking effectiveness between the two dual tasks (Table 1). Although it seemed that the mean objective score of the maintenance dual task was better than that of the manipulation dual-task, the objective scores of both dual-tasks did not show a significant difference from each other, $t(21) = 0.971$, $p = 0.343$, 95% CI [−0.03 to 0.08]. Moreover, there was no significant difference in subjective rating and $d'$ between both dual tasks, $t(21) = -1.496$, $p = 0.149$, 95% CI [−0.38 to 0.06] and $t(21) = -1.267$, $p = 0.219$, 95% CI [−0.36 to 0.08], respectively. These results indicated that the participants could not consciously perceive the masked prime of both tasks.

For the maintenance dual task, the RA of the objective test was distributed close to chance (53.2 ± 1.98%), $t(21) = 1.616$, $p = 0.121$, 95% CI [−0.01 to 0.07]. Moreover, the subjective rating was distributed close to 1 (1.24 ± 0.06). Furthermore, $d'$ values showed no significant deviation from zero (−0.06 ± 0.07), $t(21) = -0.756$, $p = 0.458$, 95% CI [−0.21 to 0.1]. Additionally, the distribution of $d'$ was normal (Kolmogorov–Smirnov = 0.101, $p = 0.2$); however, the distribution of unconscious priming under low load was not normal (Kolmogorov–Smirnov = 0.283, $p = 0.000$). Hence, logarithmic conversion was applied to the magnitude of the unconscious priming effect. The data showed that unconscious priming did not correlate with $d'$, $r(22) = -0.37$, $p = 0.09$.

For the manipulation dual task, the RA of the objective test was distributed close to chance (50.76 ± 1.88%), $t(21) = -0.403$, $p = 0.691$, 95% CI [−0.03 to 0.05]. Moreover, the subjective rating was distributed close to 1 (1.4 ± 0.09). Furthermore, $d'$ values showed no significant deviation from zero (0.08 ± 0.06), $t(21) = 1.306$, $p = 0.206$, 95% CI [−0.05 to 0.21]. Additionally, the distribution of $d'$ was normal (Kolmogorov–Smirnov = 0.118,

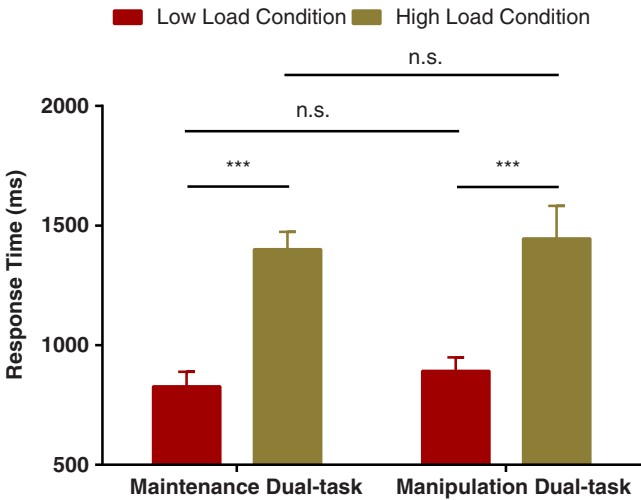

**Figure 3 Mean response times as a function of working-memory load (low *vs.* high) and experimental task type (maintenance dual task *vs.* manipulation dual task) in the working-memory task.** Red bars represent response times under the low-load condition in the maintenance dual task, and yellow bars represent response times under the high-load condition in the manipulation dual task. Error bars reflect standard errors corrected for within-participant variation. ****p* < 0.001, n.s. means no significant difference.

$p = 0.2$); however, the distribution of unconscious priming under low load was not normal (Kolmogorov–Smirnov = 0.231, $p = 0.004$). Hence, logarithmic conversion was applied to the magnitude of the unconscious priming effect. The data showed that unconscious priming did not correlate with $d'$, $r(22) = 0.055$, $p = 0.806$.

### Working-memory task

Load manipulation was efficacious for both tasks. Regarding both dual tasks, there was a main effect of load on RT (Fig. 3), $F(1,21) = 87.792$, $p < 0.000$, $\eta_p^2 = 0.807$; however, neither a main effect of task nor an interaction between task and load conditions was found, $F(1,21) = 0.298$, $p = 0.591$, $\eta_p^2 = 0.014$, $BF_{10} < 0.001$, and $F(1,21) = 0.028$, $p = 0.87$, $\eta_p^2 = 0.001$, $BF_{10} = 0.393$, respectively. In addition, a corresponding ANOVA on RA revealed that there was a main effect of load, $F(1,21) = 11.755$, $p = 0.003$, $\eta_p^2 = 0.359$; however, neither a main effect of task nor an interaction between task and load conditions was found, $F(1,21) = 0.077$, $p = 0.784$, $\eta_p^2 = 0.004$, $BF_{10} = 0.197$, and $F(1,21) = 0.19$, $p = 0.667$, $\eta_p^2 = 0.009$, $BF_{10} = 0.353$, respectively. These results indicated that there was no significant difference in load manipulation between the two tasks. Moreover, the order effects were not significant for RT and RA, $F(1,20) = 0.037$, $p = 0.85$, $\eta_p^2 = 0.002$, $F(1,20) = 1.27$, $p = 0.27$, $\eta_p^2 = 0.059$, respectively, indicating that the order manipulation was effective for the present study.

For the maintenance dual task, a load effect was observed on RT (outliers for low load: 2.93%, high: 5.03%), $t(21) = -17.734$, $p < 0.000$, suggesting that responses under low load (828 ± 61.22 ms) were significantly faster than those under high load (1,401.55 ± 72.36 ms). Additionally, a load effect was observed on RA, $t(21) = 2.921$,

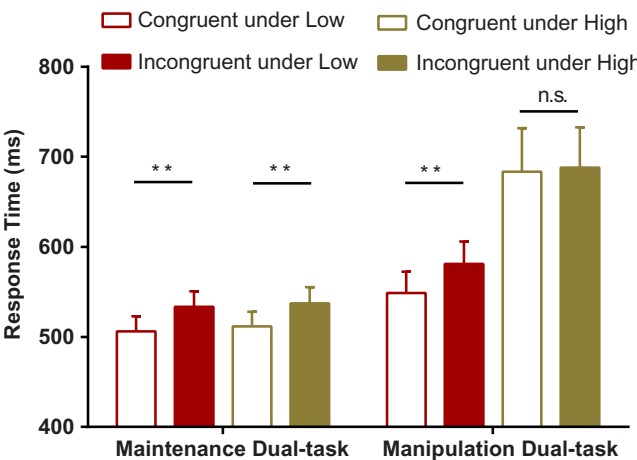

**Figure 4 Mean response times as a function of working-memory load (low *vs.* high), prime congruency (incongruent *vs.* congruent) and experimental task type (maintenance dual task *vs.* manipulation dual task) in the masked priming task.** Blank bars represent response times on congruent trials, and bars with patterns represent response times on incongruent trials. Red bars represent response times under low-load conditions, and yellow bars represent response times under high-load conditions. **\*\****p* < 0.01, n.s. means no significant difference.

*p* = 0.008, indicating that more responses under low load (97.57 ± 0.84%) were correct than those under high load (95.74 ± 0.87%).

For the manipulation dual task, a load effect was observed on RT (outliers for low load: 4.86%, high: 4.32%), *t*(21) = −4.843, *p* < 0.000, suggesting that responses under low load (892.18 ± 56.63 ms) were significantly faster than those under high load (1,446.24 ± 135.93 ms). Moreover, we also observed a load effect on RA, *t*(21) = 2.97, *p* = 0.007, indicating that more responses under low load (97.45 ± 0.45%) were correct than those under high load (95.32 ± 0.9%).

### Sandwich masking task

Regarding both dual tasks, for the RT data, the main effect for task, $F(1,21) = 7.159$, $p = 0.014$, $\eta_p^2 = 0.254$, load, $F(1,21) = 18.878$, $p < 0.000$, $\eta_p^2 = 0.473$, congruency, $F(1,21) = 14.241$, $p = 0.001$, $\eta_p^2 = 0.404$, and the two-way interaction between task and load, $F(1,21) = 15.01$, $p = 0.001$, $\eta_p^2 = 0.417$, as well as between load and congruency, $F(1,21) = 10.269$, $p = 0.004$, $\eta_p^2 = 0.328$, were significant. Moreover, no interaction between task and congruency was found, $F(1,21) = 0.73$, $p = 0.403$, $\eta_p^2 = 0.034$, $BF_{10} < 0.001$, indicating that there was no significant difference in unconscious priming between the maintenance and manipulation dual tasks. In addition, the order effects were not significant, $F(1,20) = 1.93$, $p = 0.18$, $\eta_p^2 = 0.088$, indicating that the order manipulation was effective for the present study. Most importantly, the three-way interaction between all factors was significant, $F(1,21) = 9.291$, $p = 0.006$, $\eta_p^2 = 0.307$. *Post hoc* tests yielded a significant decrease in unconscious priming only under a high load of the dual-task manipulation but not under a high load of the dual-task maintenance (Fig. 4). Furthermore, a corresponding ANOVA on RA revealed only a main effect for congruency, $F(1,21) = 5.814$, $p = 0.025$, $\eta_p^2 = 0.217$, and a two-way interaction between task and load, $F$

$(1,21) = 7.98$, $p = 0.01$, $\eta_p^2 = 0.275$. Other effects were not significant (all $Fs < 2.408$, all $ps > 0.136$). Simple effect analysis indicated that the difference in RA between both loads was significant in the dual-task manipulation but not in the dual-task maintenance.

For the maintenance dual task, a main effect of congruency was observed on RT (outliers for low: 5%, high: 4.38%), indicating that responses on incongruent trials (low: $533.39 \pm 17.32$ ms, high: $537.13 \pm 18.05$ ms) were always longer than those on congruent trials (low: $506.08 \pm 16.72$ ms, high: $511.67 \pm 16.29$ ms), $F(1,21) = 12.916$, $p = 0.002$, $\eta_p^2 = 0.381$. However, we did not observe a main effect of the load or an interaction between the load and congruency conditions on RT, $F(1,21) = 3.09$, $p = 0.093$, $\eta_p^2 = 0.128$, $BF_{10} = 0.302$ and $F(1,21) = 0.577$, $p = 0.456$, $\eta_p^2 = 0.027$, $BF_{10} = 0.283$, respectively. These results suggested that there was no significant difference in unconscious priming between low and high loads. Additionally, a main effect of congruency was observed on RA, indicating that more responses on congruent trials (low: $99.4 \pm 0.3\%$, high: $99.7 \pm 0.1\%$) were correct than those on incongruent trials (low: $99.1 \pm 0.2\%$, high: $99.3 \pm 0.2\%$), $F(1,21) = 4.797$, $p = 0.04$, $\eta_p^2 = 0.186$. However, we did not observe a main effect of the load or an interaction between the load and congruency conditions on RA, $F(1,21) = 2.469$, $p = 0.131$, $\eta_p^2 = 0.105$ and $F(1,21) = 0.104$, $p = 0.75$, $\eta_p^2 = 0.005$, respectively.

For the manipulation dual task, a main effect of congruency was observed on RT (outliers of low and high load, 2.1% and 7.37%), indicating that responses on incongruent trials (low: $580.94 \pm 24.9$ ms, high: $687.8 \pm 44.83$ ms) were always longer than those on congruent trials (low: $548.77 \pm 23.56$ ms, high: $683.26 \pm 48.36$ ms), $F(1,21) = 5.565$, $p = 0.028$, $\eta_p^2 = 0.209$. Moreover, a main effect of the load was observed on RT, $F(1,21) = 16.985$, $p < 0.000$, $\eta_p^2 = 0.447$, suggesting that responses under low loads were longer than those under high loads. Importantly, an interaction between congruency and load conditions was also observed, $F(1,21) = 10.577$, $p = 0.004$, $\eta_p^2 = 0.335$, showing that unconscious priming under low loads was significantly larger than that under high loads. Furthermore, a main effect of the load was observed on RA, indicating that more responses under low loads (congruent: $99.8 \pm 0.2\%$, incongruent: $99.8 \pm 0.1\%$) were correct than those under high loads (congruent: $99.3 \pm 0.2\%$, incongruent: $99.1 \pm 0.3\%$), $F(1,21) = 7.751$, $p = 0.011$, $\eta_p^2 = 0.27$. Additionally, neither a main effect of congruency nor an interaction between the load and congruency conditions was found on RA, $F(1,21) = 0.401$, $p = 0.534$, $\eta_p^2 = 0.019$, $F(1,21) = 0.231$, $p = 0.636$, $\eta_p^2 = 0.011$, respectively.

## DISCUSSION

The current work used an experimental method to distinguish the resource pool of the manipulation subsystem from that of the maintenance subsystem. It also investigated the relationship between working memory and unconscious processing. To our knowledge, this is the first study that focuses on distinguishing the effects of two working-memory subsystems on unconscious priming. In line with our expectations, we observed that the increase in the workload of the N-back task decreased the magnitude of the unconscious priming effect, whereas the increase in that of the Sternberg task did not decrease the magnitude of the unconscious priming effect. These observations suggested that unconscious priming shares a common resource pool with the manipulation subsystem

rather than the maintenance subsystem, which can explain past paradoxical evidence about the relationship between working memory and unconscious priming.

Our observations in the maintenance dual task are in accord with previous studies, which found that working-memory load does not affect unconscious priming (*Bodner & Stalinski, 2008*; *Perea et al., 2018*). In those studies, one masked identity priming task was interleaved with one cognitive load task; the latter required only maintaining or matching stimuli. The results showed that no decrease in unconscious priming was found, which is similar to the present findings. However, it is worth considering that there are some differences in the adopted dual-task paradigm between those studies and the present study. First, the unconscious priming elicited in the present study is different from the unconscious identity priming in those studies because the masked prime in the maintenance dual task was not identical to the target but was proportionally enlarged or narrowed relative to the target. Therefore, the unconscious priming in the present study involved more processing and required more attention resources than that in previous studies (*Kiesel, Kunde & Hoffmann, 2008*). However, we did not observe a significant decrease in unconscious priming with increasing maintenance workload, which was not due to the independence of unconscious priming on attention resources; instead it was ascribed to the independence of unconscious priming on the maintenance subsystem. Therefore, it seems inappropriate to simply summarize these findings that unconscious processing did not require resources or that working memory did not affect unconscious priming. Second, the difficulty of the dual-task paradigm in the present study was harder than that in previous studies because nine masked priming trials were intermixed with nine memory load trials in one block of the maintenance dual task. Therefore, it is conceivable that an increase in task difficulty might increase the effect of memory load on unconscious priming. However, a similar magnitude of unconscious priming between both loads was observed in the maintenance dual task, which was convergent with previous findings. Hence, the present findings in the maintenance dual task confirm and extend previous findings that unconscious priming is independent of the maintenance subsystem, and this independence is robust and generalizes across variations of the dual-task paradigm.

Our observations in the manipulation dual task are novel in that unconscious priming is impaired by an increased manipulation workload. This result is reconciled with earlier findings claiming that increasing executive attention load interferes with stimuli processing (*Bayramova et al., 2021*; *Fougnie & Marois, 2007*; *Spinks et al., 2004*). In those studies, the participants were engaged in a cognitive load task that required continuous updating or reordering of the target while an unexpected stimulus or a distractor was presented near the target. The results showed that the interference effect of the distractor dwindled or even disappeared as the executive load increased. Although both earlier findings and the present findings provide evidence of the modulation of the manipulation processes on stimuli processing, one important difference is noteworthy: in earlier studies, unexpected stimuli or distractors were shown inside of awareness, whereas in the present study, stimuli were present outside of awareness. Additionally, only one study has investigated the effect of executive attention load on unconscious processing (*Bahrami,*

*Lavie & Rees, 2007*). In that study, a foveal rapid serial visual presentation task was combined with peripherally represented stimuli that were rendered invisible by continuous flash suppression (CFS). Taking advantage of functional magnetic resonance imaging (fMRI), a decrease in the BLOD response elicited by invisible stimuli was observed when the executive attention load increased. Hence, the authors summarized that attention resources affect the processing of unconscious stimuli. However, it should be noted that the previous study observed only the neural performance of the load effect on invisible stimuli without associating any behavior performance such as unconscious priming. A recently published study reported that a previously performed induction task can affect the processing of stimuli suppressed by CFS (*Hung, Wu & Shimojo, 2020*). The authors concluded that different induction tasks led to different attention loads, which contributed to corresponding unconscious priming interference. Although that result is quite close to the present findings, two facts should be noted. First, the unconscious priming produced in that study is semantic unconscious processing, which is a relatively high-level processing and has been confirmed to require attention resources and to be modulated by mental workload (*Fabre, Lemaire & Grainger, 2007*). However, the unconscious priming produced in the present study belongs to unconscious perceptual processing, which is a relatively low-level processing and is considered an automatic process and therefore is assumed not to be affected by mental workload (*Fabre, Lemaire & Grainger, 2007*; *Perea et al., 2018*). The present study thus contradicts previous research by arguing that even low-level unconscious priming requires attention resources. Second, the so-called task-induced attention load in that study was quite different from the manipulation load in the present study; the latter is a subsystem of working memory. To our knowledge, the present study is therefore the first to directly explore the impact of the manipulation subsystem on unconscious perceptual priming at a behavioral level.

What is the mechanism by which the two subsystems show different effects on unconscious priming? Based on the present and previous studies (*Capizzi, Sanabria & Correa, 2012*; *Fabre, Lemaire & Grainger, 2007*; *Hung, Wu & Shimojo, 2020*; *Naccache, Blandin & Dehaene, 2002*; *Zanto et al., 2020*), we propose a two-pool model of attention resources to explain the moderation of attention resources on working memory and unconscious priming relationships. As summarized in Fig. 5, two relatively attention-independent resource pools are responsible for two working-memory subsystems: (1) the retention resource pool, which is not required by unconscious processes, is responsible for the maintenance subsystem, and (2) the executive resource pool, which is required by unconscious processes, is responsible for the manipulation subsystem. When increasing the workload of the maintenance subsystem, retention attention is fully depleted, while executive attention is spared. The spare resources are available to the unseen prime; thus, unconscious priming is preserved. In contrast, when increasing the workload of the manipulation subsystem, executive attention consumes full resources, which leads to insufficient resources for the masked prime; therefore, unconscious priming diminishes. In short, the mechanism by which working memory modulates unconscious processing

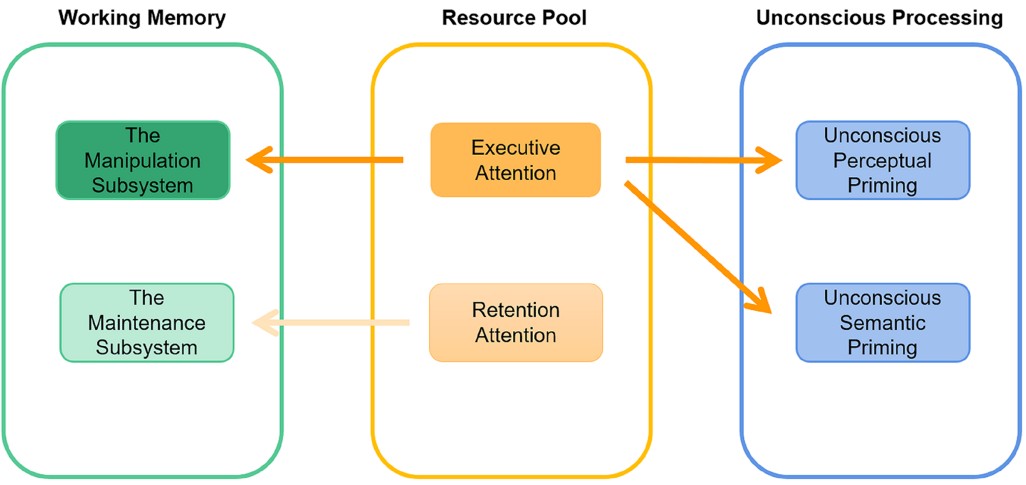

**Figure 5 Proposed two-pool model of attention resources on the relationship between working memory and unconscious priming.**

depends on the availability of the executive resource pool. These results are in agreement with those of many studies that recommend models of central executive control of unconscious priming (*Ansorge, Kunde & Kiefer, 2014*) and demonstrate that unconscious priming depends on the participants' ability to successfully allocate their attention resources to the masked prime.

The proposed two-pool model of resources required by working memory and unconscious priming can be supported by prior fMRI data. The unconscious priming elicited in the current work has been found to be associated with neural activity in the bilateral inferior and medial superior frontal gyri and corresponding regions (*Ulrich & Kiefer, 2016*), and these frontal areas are specifically associated with manipulation processes but not maintenance processes (*Tomasi et al., 2007*). Therefore, some researchers have suggested that the modulation of the manipulation subsystem on unconscious processing is related to the frontal cortex because the two kinds of processing compete for common neural resources (*Bergström & Eriksson, 2018*). However, the evidence for the neural mechanism underlying this modulatory effect is mixed. Some studies report that the superior frontal sulcal area is involved in the maintenance subsystem, while the dorsolateral prefrontal cortex is associated with the manipulation subsystem (*Glahn et al., 2002*); hence, the dorsolateral prefrontal cortex seems to be an essential neural basis for the modulation of the manipulation subsystem on unconscious priming. On the other hand, some studies report that although maintenance processes activate virtually identical neural areas with manipulation processes (*Veltman, Rombouts & Dolan, 2003*), their dynamic activation patterns are quite different (*Jolles et al., 2011*), which may account for the corresponding effects on unconscious priming. Considering that this line of research is relatively scarce, further investigations with neuroimaging techniques or other psychophysiological methods could provide deeper insights into the relation between working memory and unconscious priming at a neural-based level.

## LIMITATIONS AND FUTURE RESEARCH

The current work not only addresses the paradoxical evidence of the relation between working memory and unconscious priming but also effectively distinguishes the resource pools of two working-memory subsystems. However, the current work is not without limitations. First, to avoid the effect of the content of working memory affecting unconscious priming, we utilized geometries as the masked primes while utilizing unrelated Arabic numbers as the memory items. Based on the attentional sensitization model, however, the previously performed induction task affects subsequent unconscious priming. In this proposed model, a perceptual induction task facilitates perceptual unconscious priming, whereas a semantic induction task dampens perceptual unconscious priming (*Kiefer, 2019*). Hence, it seems rational that the decline in unconscious priming with the loaded condition observed in the manipulation dual task stemmed from the task set rather than the engagement of the manipulation processes. Nonetheless, it should be pointed out that the stimuli we utilized in the induction task (memory task) were the same between both dual tasks; therefore, if the interference effect exists, the interference effect of both tasks on unconscious priming should be identical. Since both induction tasks contained the maintenance processes but only the manipulation dual task contained the manipulation processes, any difference in unconscious performance should originate at the engagement of the manipulation subsystem. On the other hand, additional variables, such as the effect of the type of induction task on the modulation of working memory on unconscious priming, deserve more attention. Thus, future research should vary the type of induction task as well as the type of unconscious priming to explore more possibilities associated with the modulation of working memory on unconscious priming. Moreover, the present data evaluated the effect of working memory on unconscious perceptual priming but not other types of unconscious processing (*e.g.*, unconscious semantic priming). Hence, our hypothesized model needs corroborating by additional studies. Additionally, we only recruited a relatively small sample of participants; hence, we cannot exclude the impact of individual differences on our results. Future studies can examine the present results with a larger sample size. Finally, although this paper demonstrates the effect of working memory on unconscious priming at the behavioral level, the neural mechanism underlying this phenomenon remains unclear and is worth exploring in the future.

## CONCLUSION

The present article tested the effect of two working-memory subsystems on unconscious priming and discriminated different resource pools among the two subsystems. To this end, participants were engaged in the sandwich masking task while storing or updating the memory items in two experimental tasks. The results indicated that unconscious priming is affected by the manipulation workload rather than the maintenance workload. We then proposed a two-pool attention resource model to explain the relationship between working memory and unconscious priming and provided some neuro-based evidence to support the proposed model. Despite a growing consensus in the literature that holds that unconscious processing depends on attention resources, the current study is the

first to show that unconscious priming shares a common resource pool with the manipulation subsystem. By introducing unconscious priming, we are capable of distinguishing the two resource pools among the two subsystems: one is needed for unconscious priming, while the other is not. Additionally, by distinguishing the two resource pools, we are capable of explaining previous mixed findings regarding the relationship between working memory and unconscious priming. Some studies investigate the relationship between maintenance processes and unconscious priming, while some studies investigate the relationship between manipulation processes and unconscious priming.

## ACKNOWLEDGEMENTS

We thank Liming Deng from the Royal Melbourne Institute of Technology for his help with statistical consultation. We also thank Sebastian Wilson from Missouri University of Science and Technology for his help with English language editing.

### Funding

This research was supported by the Innovative Research Group Project of the National Natural Science Foundation of China (Grant Numbers: 31971023). The funders had no role in study design, data collection and analysis, decision to publish, or preparation of the manuscript.

### Grant Disclosures

The following grant information was disclosed by the authors:
National Natural Science Foundation of China: 31971023.

### Competing Interests

The authors declare that they have no competing interests.

### Author Contributions

- Xuechen Mao conceived and designed the experiments, performed the experiments, analyzed the data, prepared figures and/or tables, authored or reviewed drafts of the paper, and approved the final draft.
- Anmin Li conceived and designed the experiments, authored or reviewed drafts of the paper, and approved the final draft.

### Human Ethics

The following information was supplied relating to ethical approvals (*i.e.*, approving body and any reference numbers):

Shanghai University of Sport Ethics Committee approval to carry out the study within its facilities (No. 102772021RT020).

## Data Availability

The raw measurements are available in the Supplemental File.

## Supplemental Information

Supplemental information for this article can be found online at http://dx.doi.org/10.7717/peerj.13312#supplemental-information.

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
