# Peer review of "Unconscious priming shares a common resource pool with the manipulation subsystem"

_PeerJ, doi:10.7717/peerj.13312_

## Round 0.1 · original submission · Major Revisions

I feel very fortunate that I was able to receive two very well-articulated and insightful reviews from experts on this topic. Both reviewers find merit in your topic and find your study to be well rationalized and rigorous. However, both note a number of important improvements that are needed before the study will be publishable. I concur with their assessment and agree with their astute suggestions. Of note, we all find your use of the word “task” to be imprecise and confusing. However, I generally found the paper to be very clear and readable and congratulate you on a nice, concise study.

I think you should examine order effects even if order was counterbalanced.

Both reviewers point to some issues with the writing. Given that I am asking for major revisions, I am not going to do extensive line by line editing here, but please note the following:

Line 20 of the abstract needs revision.

Line 74 delete “that”

Line 94, use “same” instead of congruent outside of parentheses to align with “different” in the second clause and avoid circularity. Then you won’t need the subsequent sentence.

Be sure to define what you mean by sandwich task for less expert readers.

Explain why you chose a within subjects design. You can emphasize the increased power and control. Justify the small sample size.

Be clear if all participants received a unique ordering of stimuli in each task.

Report confidence intervals for analyses where possible.

Line 461, do you really mean “decline” or do you mean lower level of compared to the other task?

Line 473 move the “only” to before “unconscious perceptual priming.”

Figures 3 -5 are odd ways to depict the data with participants on the x axis. I think a table showing the magnitude and direction of difference between performance on the two tasks would be more useful. For Figure 4, subjective rating of what exactly?

I think Figure 7 should be a bar graph. You are just plotting four average scores, not continuous data, correct?

Reviewer 1 ·

Basic reporting

The manuscript aims to distinguish the resource pools of the subordinate working memory systems, namely the maintenance and manipulation subsystems. The authors further employed a masked priming paradigm in a dual-task program to investigate the relationship between these subsystems and unconscious priming. They conclude that the manipulation subsystem interferes with unconscious priming and propose a two-pool attention resource model.
The manuscript provides a clear account of the motivation and reasoning of the experiments. The hypotheses formulated in the introduction are supported and derived from the literature that is provided. Overall, the manuscript addresses an interesting issue. However, substantial additional work is required to enhance readability and interpretation, especially concerning the literature review and descriptions of the design. Therefore, I conclude that the present manuscript is not ready for publication.

Overall, the English language should be improved (e.g. lines 63, 67, 80, 212f, 355-359). The current phrasing makes comprehension difficult, especially in sections concerned with the design and procedure of the experiments. I suggest that you improve the descriptions in lines 192-226. Statistical abbreviations should be italicized when appropriate. Moreover, the results section could be improved in terms of readability.

The authors provide an overview of the relevant literature, but the introduction requires more detail. Cognitive load theories, response priming, masking, and the employed tasks in general should be discussed in light of relevant publications (see for example Schmidt, Haberkamp & Schmidt, 2011; Klotz & Neumann, 1999; Klotz & Wolff,1995; Eimer & Schlaghecken, 1998; Sternberg, 1969, 1975). Extensive additions should be made regarding the concept of consciousness.

“Although previous findings are mixed, further analysis indicates that the load manipulation tasks utilized in those studies seem to involve different working-memory subsystems; thus, this could account for the paradoxical findings.” (line 81ff). Which tasks were utilized?

The structure of the article conforms to the required journal article conventions.
Raw data was not provided, but processed and averaged data, excluding subjects omitted from further analysis.
Overall, figures should be provided in higher resolution. Indication of significance in Figure 7 should be clearer (which effects are tested).

References:
Eimer, M., & Schlaghecken, F. (1998). Effects of masked stimuli on motor activation: Behavioral and electrophysiological evidence. Journal of Experimental Psychology: Human Perception and Performance, 24(6), 1737–1747. https://doi.org/10.1037//0096-1523.24.6.1737

Klotz, W., & Neumann, O. (1999). Motor activation without conscious discrimination in metacontrast masking. Journal of Experimental Psychology. Human Perception and Performance, 25(4), 976–992. https://doi.org/10.1037/0096-1523.25.4.976

Klotz, W., & Wolff, P. (1995). The effect of a masked stimulus on the response to the masking stimulus. Psychological Research, 58(2), 92–101. https://doi.org/10.1007/BF00571098

Sternberg, S. (1969). Memory-scanning: Mental processes revealed by reaction-time experiments. American scientist, 57(4), 421-457.

Sternberg, S. (1975). Memory Scanning: New Findings and Current Controversies. Quarterly Journal of Experimental Psychology, 27(1), 1–32. https://doi.org/10.1080/14640747508400459

Experimental design

The submission defines the research question and identifies a relevant research gap. Manipulation of the maintenance and manipulation subsystems required in the secondary task allows investigating their respective effects on the primary priming task. Through this, the subsystem involved in unconscious priming can be identified.
Overall, the experiment has been well-designed.
However, to ensure replication of the experiment or to make comprehension easier, some clarifications are needed:
1) It is unclear to me how many trials of each condition were employed in exactly how many blocks across how many sessions. Because of this, it is difficult to assess whether enough data was acquired to make decisions on significance.
To reduce confusion, I suggest using the term ‘task’ only when referring to the two variations tested. Otherwise, you should follow the hierarchical order trial, block, session. Most importantly, avoid ambiguity when describing the procedure in lines 154-165.
2) As mentioned in the previous section, extensive work is required in lines 192-226. It is especially difficult to follow the description of the high-load trials in the second paragraph. Specifically, you should be clearer when describing the change from memory probe to memory item. As I understand, the initial first memory item leaves the set and is replaced by the current memory probe. In the next trial, the previously second item is probed, and so on.
3) The authors fail to thoroughly explain the employed N-back task. Typically, the N-back task requires a stepwise increase of N (first 1-back, followed by 2-back, then 3-back). Further, they never specifically state that they are using the 3-back as the high-load condition, and the 1-back as the low-load condition.
4) It appears that the ethics approval of another experiment was submitted.

Validity of the findings

Data was only provided for the subjects that were analyzed. Excluded subjects are missing. Moreover, the provided data is already processed and averaged.
Overall, I commend the authors for linking their findings to the original research questions. However, in order to make a more compelling case, the following issues should be addressed:

1) Exclusion criteria for subjects need to be justified and explained in more detail. For example, it is stated that subjects with accuracy significantly larger than chance were excluded. How exactly was this determined?
2) It is unclear how the authors deal with the problem of multiple comparisons. Were corrections used? If so, which correction? Either way, the decision needs to be justified.
3) The authors should be clearer in their definition of ‘unconscious priming’ in the reporting of their results. For example, it is not clear what they refer to exactly when writing “however, the distribution of unconscious priming under low load was not normal” (line 263f). For example, the priming effect can both be measured in differences in accuracy and reaction time.

Additional comments

The manuscript should also be checked for typos and spelling. Sometimes it seems that in the process of revising the manuscript multiple times, parts of sentences have been lost (e.g., lines 73-75).

Reviewer 2 ·

Basic reporting

1. The majority of the manuscript is clear and easy to follow. There are some small inaccuracies in the English sometimes (e.g., prepositions, time), but it should be possible to resolve them easily. What was, however, very confusing to me was the usage of the word task/Task.
First, I would suggest that the authors pick more self-explanatory names than Task 1 and Task 2 (e.g. maintenance task and manipulation task or Sternberg task and n-back task would be more informative and one would not always need to keep in mind what 1 and 2 were referring to again). In the Abstract, it appears that the authors themselves confused the tasks, as they wrote that the working memory load did and did not decrease unconscious priming in Task 2 which is clearly contradictory (Results section of the Abstract).
Second, the word task was apparently used in several senses - the task to perform (Sternberg vs. n-back), a phase of the experiment? (practice task, main task, ...), as well as an equivalent of trial? (360 Sternberg tasks). This confusion is creating misunderstandings especially in the first paragraph of "Procedure". I highly recommend that the authors optimize their terminology in this paragraph and transfer the corresponding terms to the rest of the manuscript, so that the terminology is consistent and sufficiently distinct. In terms of consistent terminology, I also wanted to point out that the authors switch between retention attention resource pool and retention resource pool and between executive attention resource pool and executive resource pool. This should also be changed, so that the terms remain the same throughout the manuscript.

2. When referring to perspectives on automaticity in the 4th paragraph of the introduction, the authors should also cite at least one review article that describes different concepts and theories of automaticity (e.g., Moors & De Houwer, 2006).

Experimental design

The research question was developed and motivated very well and is very meaningful. However, there are some aspects of the methods description and the analysis that should be optimized before publication.

1. The authors should report a sample size estimation, so that the reader can evaluate whether their study was sufficiently powered to detect the effects of interest. As it appears that the authors did not compute a sample size a priori, they should at least elaborate on the power they were able to achieve for the comparisons of interest given the sample actually included in the analyses. This is especially relevant as the most essential effect the authors needed to find was a three-way interaction and a relatively large effect size is necessary to observe such a three-way interaction with sufficient power with a sample of only 22 participants. Therefore, more information on the authors' sample size considerations is essential.

2. Building on the previous aspect. A non-significant two-way interaction for Task 1 does not constitute strong evidence against an interaction especially given the small sample size. I think the authors' rationale and findings are very compelling, but they could improve their statistical foundation by additionally reporting corresponding Bayesian results, so that evidence against an interaction for Task 1 can also be assessed. For instance, the free JASP software provides easy opportunities for conducting such a parallel Bayesian analysis.

3. The high load n-back task did not become entirely clear to me. Based on the authors description, it seemed as if the first memory item was the only one relevant for later comparison. Then, however, the task would not have involved an increased load, in my opinion. Maybe I misunderstood something here, but in either case, the authors should explain the n-back task in a bit more detail to prevent misunderstandings.

4. For the sandwich masking task, it appears that the authors are reporting both ANOVAs per Task1/2 and an overall ANOVA with 3 factors. Only the 3-factor ANOVA should be reported as it includes all findings of the respective Tasks 1 and 2. Moreover, sometimes the authors only write that there was/ was no interaction. They should always refer to the respective interaction mentioning the involved factors to avoid misunderstandings. Moreover, as there are multiple analyses on the masked prime discrimination test, the working memory load induction, and the sandwich masking task, it would be easier to follow for the reader if the analyses per part were reported at the beginning of the result description of the respective part rather than in a - at present a bit hard to disentangle - joint paragraph at the beginning of the results section

5. The authors report that RTs that differed by more than 2 standard deviations from the mean were excluded. However, they did not specify whether the overall sample mean, the participant mean, or the individual cell mean was the reference. This information should be added. Furthermore, if the authors did not use the individual cell mean (i.e., the mean per participant and condition) as the reference, they should adapt their analyses to do so. Using the sample or participant mean distorts the data as fast/slow participants' data is more likely excluded or trials from a condition with faster/slower responses are more likely excluded. Thus, the individual cell means should be used as reference for computing the standard deviations.

6. In terms of terminology, the authors report that "responses were more correct". Responses can only be correct or erroneous. The phrasing should thus be changed to "fewer/more responses were correct".

Validity of the findings

The rationale of the study is clearly stated, timely, and interesting to the research community.

1. In their discussion, the authors state that "spare resources spill over involuntarily". I might have misinterpreted this sentence, but to me it seemed to contradict their description of their model elsewhere as well as its graphical illustration. I would suggest to change the phrasing of this sentence to avoid misunderstandings.

2. Furthermore, the atuhros write that "unconscious perceptual processing .... is considered an automatic process and is not affected by mental workload ...". This should be adapted to "... is considered an automatic process and therefore is assumed not to be affected by mental workload...". Otherwise, the factual statement at the end of the sentence would be contradicting other parts of the discussion.

---

## Round 0.2 · Minor Revisions

I was very lucky to have both expert reviewers assess your revision. While they are happy with most aspects of your revision, they note some lingering issues with analyses and the sharing of raw data, which must be addressed before I can accept the paper.

Reviewer 1 ·

Basic reporting

The language has improved substantially. I especially commend the extensions with regard to the relevant literature.
If the following minor changes/corrections are implemented, I recommend the publication of this article.


I noticed a couple of minor issues still:

Line 93: AAAA are vowels
Line 95: it should be BOLD response
Line 100: COMPARED TO low loads
Line 138: a lower level of unconscious processES
Line 205-208: I think this sentence could be a little clearer with a small change:
and 6 blocks of either of the dual tasks (maintenance: nine sandwich masking trials were intermixed with nine modulated Sternberg trials per block; manipulation: nine sandwich masking trials were intermixed with nine N-back trials per block)
Lines 215ff: I think trial was more appropriate than block here

Experimental design

The description of the task is a lot clearer now.

Validity of the findings

Original review:
2) It is unclear how the authors deal with the problem of multiple comparisons. Were corrections used? If so, which correction? Either way, the decision needs to be justified.
Response:
2) Yes, Greenhouse–Geisser corrections were used. We added corresponding statements in the new manuscript “Importantly, Greenhouse–Geisser corrections were applied when sphericity assumptions were violated, although these corrections did not change any inferences” (Lines 539-541 in the Analysis section).

I commend the inclusion of Greenhouse-Geisser corrections. However, as the authors correctly state, Greenhouse-Geisser is a correction for the violation of the sphericity assumption.
My original concern regarded multiple comparisons. Given the large quantity of t-Tests, how did you deal with the multiple comparison issue? Similarly, which post-hoc corrections were used, e.g. line 378 (Bonferroni, LSD, etc.)?

Additional comments

Figure 1, Figure 2, and Figure 5 seem to be of low resolution still.

The ethics approval document (English translation) provided, appears to still be concerned with another experiment.

Reviewer 2 ·

Basic reporting

My prior concerns regarding basic report have effectively been addressed.

In the new version of the manuscript, the authors now refer to "the inhibition effect from the incongruent trials" in line 111. They did not introduce or explain this term and the reader can only guess what exactly they refer to. I therefore suggest that the authors adjust the phrasing of the corresponding sentence.

Experimental design

1. I thank the authors for adding the requested Bayesian analyses which provide a clear-cut picture for the corresponding effects. Yet, now the authors offer more information on sandwich masking in lines 374-383. There, they also try to draw the conclusion that there is no difference between conditions. To do so, corresponding Bayesian anaylses would again be required. This also applies to other sections of the results where the authors state that there was no difference between conditions. Such a conclusion can only be drawn after a corresponding Bayesian analysis. Otherwise, it can only be stated that there was no significant difference, but not that there was no difference. Furthermore, there and elsewhere the authors now mention a 90% confidence interval. Why is a 90% confidence interval instead of the common 95% confidence interval reported?

2. However, I am not yet fully satisfied with the information on sample size estimation the authors provided. An ANOVA sample size estimation should always refer to the assumed (mean/singluar) effect size for the respective effect observed in one or multiple prior studies that the authors use as basis or postulated by the authors due to theoretical considerations. As the assume effect size is the essential basis of any sample size estimation, please add this information.

3. As requested by the editor, the authors added an analysis considering order. However, order was added as a covariate rather than a between-subject factor. Why was this done?

Validity of the findings

.

Additional comments

As Reviewer 1 already pointed out explicity in the first round of reviews, the authors did not make their raw data available so far. They only provided an aggregated version which contained the means per condition for only the participants included in the analyses. Furthermore, the authors did not react to R1's comment regarding raw data. In my opinion, the manuscript should not be accepted before the raw data is made available as required by PeerJ.

---

## Round 0.3 · accepted · Accept

Thank you for being responsive to the reviewers' comments in the last round of review. I commend you on a nice study and your careful attention to the recommendations of the reviewers.